# InBRwSANet: Self-attention based parallel inverted residual bottleneck architecture for human action recognition in smart cities

Yasir Khan Jadoon[1], Muhammad Attique Khan[2]*, Yasir Noman Khalid[1], Jamel Baili[3], Nebojsa Bacanin[4], MinKyung Hong[5], Yunyoung Nam[6]*

**1** Department of Computer Engineering, HITEC University, Taxila, Pakistan, **2** Department of AI, College of Computer Engineering and Science, Prince Mohammad Bin Fahd University, Al Khobar, Saudi Arabia, **3** Department of Computer Engineering, College of Computer Science, King Khalid University, Abha, Saudi Arabia, **4** Faculty of Informatics and Computing, Singidunum University, Belgrade, Serbia, **5** Emotional and Intelligence Child Care System Convergence Research Center, Soonchunhyang University, Asan, Republic of Korea, **6** Department of Computer Science and Engineering, Soonchunhyang University, Asan, Republic of Korea

\* ynam@sch.ac.kr (YN); attique.khan@ieee.org (MAK)

## Abstract

Human Action Recognition (HAR) has grown significantly because of its many uses, including real-time surveillance and human-computer interaction. Various variations in routine human actions make the recognition process of action more difficult. In this paper, we proposed a novel deep learning architecture known as Inverted Bottleneck Residual with Self-Attention (InBRwSA). The proposed architecture is based on two different modules. In the first module, 6-parallel inverted bottleneck residual blocks are designed, and each block is connected with a skip connection. These blocks aim to learn complex human actions in many convolutional layers. After that, the second module is designed based on the self-attention mechanism. The learned weights of the first module are passed to self-attention, extract the most essential features, and can easily discriminate complex human actions. The proposed architecture is trained on the selected datasets, whereas the hyperparameters are chosen using the particle swarm optimization (PSO) algorithm. The trained model is employed in the testing phase for the feature extraction from the self-attention layer and passed to the shallow wide neural network classifier for the final classification. The HMDB51 and UCF 101 are frequently used as action recognition standard datasets. These datasets are chosen to allow for meaningful comparison with earlier research. UCF101 dataset has a wide range of activity classes, and HMDB51 has varied real-world behaviors. These features test the generalizability and flexibility of the presented model. Moreover, these datasets define the evaluation scope within a particular domain and guarantee relevance to real-world circumstances. The proposed technique is tested on both datasets, and accuracies of 78.80% and 91.80% were achieved, respectively. The ablation study demonstrated that a margin of error value of $70.1338 \pm 3.053$

**Data availability statement:** The datasets used in this work are publically available for the research purposes such as HMDB51 (https://serre-lab.clps.brown.edu/resource/hmdb-a-large-human-motion-database/) and UCF101 (https://www.crcv.ucf.edu/data/UCF101.php).

**Funding:** The authors extend their appreciation to the Deanship of Research and Graduate Studies at King Khalid University for funding this work through Large Research Project under grant number RGP.2/275/46 (Awarded to YKJ). This work was supported by the National Research Foundation of Korea(NRF) grant funded by the Korea government(MSIT) (No. RS-2023-00218176) and the Soonchunhyang University Research Fund (Awarded to YKJ).

**Competing interests:** All authors declared no conflict of interest in this work.

($\pm$4.35%) and 82.7813 $\pm$ 2.852 ($\pm$3.45%) for the confidence level 95%, $1.960\sigma\bar{x}$ is obtained for HMDB51 and UCF datasets respectively. The training time for the highest accuracy for HDMB51 and UCF101 is 134.09 and 252.10 seconds, respectively. The proposed architecture is compared with several pre-trained deep models and state-of-the-art (SOTA) existing techniques. Based on the results, the proposed architecture outperformed existing techniques.

## 1. Introduction

Human Action Recognition (HAR) has become an essential area for research in pattern recognition and computer vision in recent decades, and it has significantly enhanced video interpretation [1]. Video surveillance, robotics, patient monitoring systems, human-computer interface, and pedestrian detection are some of the fields in which HAR is usually employed [2]. This procedure involves identifying movements inside video frames, such as walking, running, punching, leaping, and playing, to enable automated activity recognition systems [3]. HAR encompasses a range of technologies, including wearable sensor-based, wireless sensor network-based, and video-based techniques [4]. Due to its high action detection accuracy and ease of organization, video-based HAR has gathered significant attention and is widely utilized in various industrial applications [5].

Accurately identifying human actions in videos is still challenging because of several variables, including variances within and between action classes, changing illumination, environmental circumstances, and camera angles [6]. Prior studies entirely rely upon handcrafted feature technologies like histogram-oriented gradients (HOG) and histogram of optical flow (HOF) using 2D camera video frames [7]. Usually, these techniques examine how human body parts appear and move within video frames; however, they do not consider the three-dimensional setting of action frames within a video frame [8]. As such, employing single-modality RGB/RGB-D video frames for HAR is insufficient to tackle the practical issues mentioned earlier [9]. This scenario has dramatically improved due to recent improvements in depth-sensing camera technology, which provides detailed 3D information regarding moving human body parts and changes in postural positions [10]. This makes it possible to calculate joint angles and distances, creating a distance vector for every video frame [11].

The traditional approach for human analysis, segmentation, and action classification separates human silhouettes from dark and noisy backgrounds, following the movements of body parts and classifying the activities people execute [12]. Traditional handcrafted features are typically employed [13]. Convolutional neural networks (CNNs) have been demonstrated in recent research to improve significantly recognition performance [14]. The computer vision community has taken notice of CNNs due to their enhanced performance in a variety of applications, including biometrics [15], surveillance [16], medical imaging [17], object classification [18], and agriculture [19]. CNN has a greater capacity to handle large datasets [20]. Transfer learning is used in several CNN-based techniques, using pre-trained models such as EfficentNet, ResNet, and others [21].

A vision-based recognition system poses a solution that is based on non-contact. The detection is considered scalable and can be utilized for real-world applications. However, physical attachment is necessary in sensor-based approaches, which can be invasive. Spatial and temporal feature information can be attained using a vision-based approach. It makes it possible to classify human actions and related contextual information. DCNN-based methods classify actions and activities under various circumstances using a vision-based approach. The current research further highlights the potential of vision-based approaches. Vision-based approaches have proved the effectiveness of neural network-based dual-stream architectures in addressing complex spatiotemporal features and low-lighting issues [22]. Several novel frameworks, like slow-fast tubelet (SFT), perform very well in lesser light and infrared conditions [23]. The challenges specific to the domain of action recognition during sports activities are highlighted. Domain-specific action recognition needs more specialized datasets [24]. These findings motivate the development of a customized DCNN for action recognition by stressing the flexibility and elasticity of vision-based methods.

Newly customized DCNN presents an efficient and effective way to recognize human actions. Its applicability is widespread due to the usage of benchmark datasets (i.e., HMDB51 and UCF101). The proposed approach does not concentrate on domain-specific datasets like [24]; hence, it guarantees wider comparison and pertinence. The proposed architecture can handle RGB-specific recognition of actions without relying on specialized preprocessing, as in the case of infrared-based preprocessing [23]. A single-stream method is proposed in the proposed approach, and it performs better on HMDB51, whereas the method by [22] is a dual-stream method, and it works better on the UCF101 dataset. The single-stream proposed DCNN compromises efficiency and accuracy, making it suitable for many tasks without extra complexity. The transformer-based network, in combination with MobileNetV3 and integration of self-attention, is used to overcome the limitations of computation in consumer electronics [25]. In the proposed approach, a general-purpose action recognition methodology is devised. The methodology takes care of the extraction of essential features and obtains competitive performance without using complex transformers. Industrial HAR issues are resolved using squeeze bottleneck attention block (SBAB) and sequential temporal convolution network (STCN) [26]. At the same time, the proposed customized DCNN provides a simplified and effective solution for RGB-based action recognition on HMDB51 and UCF101. The method maintains a balance between efficiency and computing efficacy without losing accuracy. So, the customized DCNN is a good choice for benchmark datasets and real-world applications.

Pretrained CNN models are generalized models; however, considering the complex nature of human action recognition systems, there is an immense need to develop a customized deep neural network model. The pre-trained models contain many learning parameters, such as learnability; therefore, designing a model that can work on complex and accurately recognize similar actions such as running, jogging, walking, and many more is essential. In addition, several complex human actions (i.e., push, punch, push up, pull) make the action recognition task difficult and, in the end, increase the loss value. We proposed a novel deep-learning architecture for accurate HAR using the video frames in this work. The designed model can easily extract features of the given actions at different levels and then easily differentiate them from similar actions. Our significant contributions are:

- Proposed a new architecture for HAR called Inverted Bottleneck Residual with Self-Attention (InBRwSA)

- Proposed a multi-level Six Parallel Inverted Residual Bottleneck architecture that can capture feature details of the given sequences at low and high levels.

- Proposed an attention-based mechanism at the end of the proposed architecture to capture attention-oriented feature maps from the video sequences.

- A detailed ablation study was conducted, and we compared the proposed architecture with state-of-the-art pre-trained models such as ResNet, Mobilenet, Inception-ResNet, and DenseNet201.

## 2. Literature review

Much work has been done in the literature for HAR using machine learning (ML) and deep learning (DL) techniques. The ML techniques mostly focused on the traditional methods such as shape features, contour features, local features, and geometric features; however, these features are not performed enough for large-scale complex datasets such as HMDB51 and UCF101. Therefore, deep learning techniques have been widely employed in computer vision in the last decade for several applications. HAR is one application that attracted several researchers in this domain and introduced several solutions to reduce recognition loss.

Dastbaravardeh et al. [27] presented an approach in which a combined methodology using autoencoders, convolutional neural network (CNN), channel attention mechanism(CAM), and low-size/low resolution is used to improve the visibility of actions in video frames. The authors used the HMDB 51 dataset to validate the presented approach and obtained 77.29% accuracy for action recognition. Several future directions were suggested, including fine-tuning the neural network model and using multimodal approaches with real-time implementation. In another approach, Amin et al. [28] presented a practical and well-optimized CNN-based system to process real-time data streams obtained from the optical sensor of a mobile surveillance system. First, a CNN model that had already been trained was used to extract frame-level deep features. A deep autoencoder (DAE) was tuned and used to understand how the actions in the surveillance stream changed over time.

Moreover, quadratic SVM was trained to classify human activities. This approach achieves an accuracy of 70.30% using the HMDB51 dataset. The authors intended to provide multiview action detection for dynamic situations and to evaluate and track numerous activities in video streams as a future task. Another HAR technique is presented by Joudaki et al. [29], who built CNN based on deep belief networks (DBN). It contributed to DBNs' capacity to analyze and interpret two-dimensional video frames and comprehend time through recursive implementation. Using constrained Boltzmann machines, the presented technique could comprehend long-term temporal motions based on recursive implementation and short-term temporal concepts. It was tested on the HMDB51 dataset and obtained an accuracy of 74.28%. As a future work, the authors suggested hybrid methods for improved accuracy. Abdorreza et al. [30] used preprocessing stages such as shot detection, cancelation of camera movement, and correct human location specification for improved action recognition. Utilizing preprocessed features for deep learning classification, this approach is integrated with no computational overhead. When tested on a deep learning-based HAR system, it produced an accuracy of 71.13% on the HMDB51 dataset.

Yang et al. [31] described a deep learning network based on fusion of spatio-temporal features (FSTFN). It was presented to address the problem of existing deep learning networks not adequately extracting and fusing spatiotemporal information in action identification tasks, leading to low accuracy. The presented model extracted and fused temporal and space information using two networks: CNNs (Convolutional Neural Networks) and LSTMs (Long Short-Term Memory). Large-scale video frame data was processed using multi-segment input, which resolved the issue of long-term reliance and increased prediction accuracy. The attention mechanism also increased the visual subjects' weight in the network. An accuracy of 65.90 is achieved on the HMDB51 dataset. In addition to being investigated for videos, contrastive learning has almost wholly bridged the gap between supervised and self-supervised image representation learning.

Nevertheless, previous research did not examine how differentiating characteristics worked across the temporal dimension. Dave et al. [32] created a new temporal contrastive learning framework with two novel losses. While the global-local temporal contrastive increased temporal variety by discriminating across timesteps of an input frame's feature map, the local–local temporal contrastive loss distinguished between non-overlapping frames from the same video. Across numerous datasets and backbones, the presented system greatly outperformed the state-of-the-art results in various video understanding tasks, including action detection and nearest-neighbor video retrieval. Shen et al. [33] suggested an architecture named the 2D Progressive Fusion (2DPF) Module, which was added after the 2D backbone CNN layers, to

solve issues of network convergence and recognition accuracy. To increase recognition accuracy and convergence, the presented 2DPF architecture shows better performance. Benchmark dataset UCF101 was used for the validation, and an accuracy of 79.60% was obtained.

Ahmad et al. [34] presented a CNN technique based on the Bidirectional-gated recurrent units (Bi-GRU) to present accurate human activity recognition. Authors used CNNs to extract deep features from video frames to improve performance and lower computational complexity. The extracted features are further learned in both forward and backward temporal dynamics using Bi-GRU. The presented method was tested on the UCF101 dataset and obtained an accuracy of 91.79%. Wang et al. [35] suggested an approach called Adaptive Contrastive Learning (ACL), which used adaptive selection to pick positive-negative samples for contrastive learning and evaluated the confidence of unlabelled samples. They used the UCF101 dataset and obtained 88.60%.

## 3. Mathematical formulation of deep modules

This section presents the mathematical formulations of the deep learning modules, such as residual, bottleneck, and inverted residual.

**Residual module:** A residual block implements identity mapping using the Equation 1:

$$y_{m+1} = y_m + R(y_m, V_m) \tag{1}$$

Here, $y_{m+1}$ and $y_m$ represent the input and output for the $m^{th}$ unit in the neural network. R. Parameter represents a residual function for the function are $y_m$, $V_m$. In a residual neural network, residual blocks are stacked sequentially. Fig 1 represents the behavior of the residual network by implementing identity mapping [36].

**Bottleneck module:** Utilizing linear bottlenecks to compress input features before extending them back to a higher dimension helps the model to operate more efficiently by reducing the dimensionality of the data. The behaviour of linear bottleneck is mathematically formulated below [37]. In an expansion phase, a pointwise convolution is applied using $1 \times 1$ filters. Mathematically, it is defined by Equation 2.

$$y_{\exp} = \alpha(y * W_{\exp}) \tag{2}$$

At the next step, a depth-wise separable convolution is applied to capture spatial features, as given in Equation 3.

$$y_{dep} = \alpha(y_{\exp} *_{dep} *W_{dep}) \tag{3}$$

In the final step, a projection is represented using pointwise convolution, as defined by Equation 4.

$$y_{proj} = y_{dep} * W_{proj} \tag{4}$$

In the above Equations 2–4 $y$ represents the input tensor, $y_{\exp}$ represents the weight vector for pointwise convolution, the activation function is denoted by $\alpha$, depth-wise convolution operation is represented by $*_{dep}$, $W_{dep}$ represents depth-wise filters, $W_{proj}$ is a weight vector for pointwise operation. Visually, it is illustrated in Fig 2.

**Inverted residual module:** The expansion, depth-wise convolution, and projection phases are followed in the same manner as mentioned in Equations 2–4. If the input and output dimensions match, a residual connection is added.

$$Z = y + y_{proj} \tag{5}$$

If the input and output dimensions do not match, the resultant will be $y_{proj}$ [37].

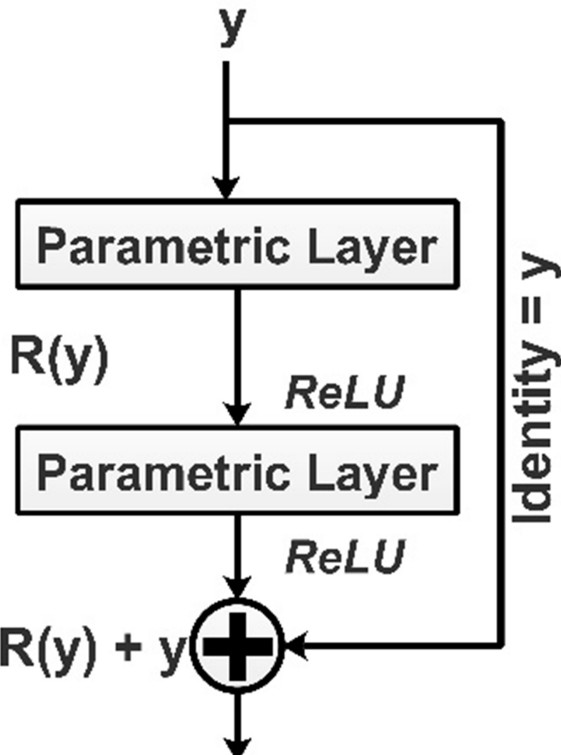

**Fig 1. Residual behavior.**

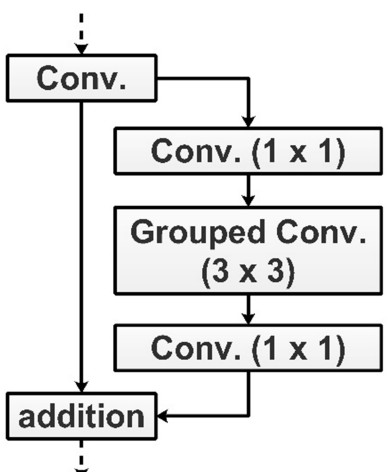

**Fig 2. Inverted residual bottleneck block.**

## 4. Proposed architecture

This section presents the proposed HAR architecture. The overall flow of the methodology is represented in the algorithmic formulation below.

## Algorithm: Proposed methodology

```
Step 1: Selection of HMDB51 and UCF101 datasets
Step 2: Customized DCNN with 6 number of inverted residual blocks
Step 3: implementation of the self-attention mechanism.
Step 4: The model is trained on selected datasets
Step 5: Features are extracted.
Step 6: Multiple classifiers are used to classify features into relevant classes.
Step 7: Performance measures are used to validate the proposed model.
```

The proposed architecture, named Inverted Bottleneck Residual with Self-Attention (InBRwSA), consists of 211 layers with 8.7 million learnable parameters. The model starts with a convolutional layer and is later on followed by six blocks in series in which each block consists of 6 parallel inverted residual bottleneck blocks. After the sixth block, a global average pool layer is used, and data is then flattened to make its input for the self-attention layer. Finally, fully connected and Softmax layers were included, which ended with a classification output layer. The proposed architecture is trained, and later features are extracted to test the final classification layer. Fig 3 shows the detailed proposed architecture for HAR. A detailed description of each step is given in the subsections below.

### 4.1. Datasets

This work utilizes two datasets for the experimental process: HMDB51 [38] and UCF101[39]. The HMDB51 dataset contains 51 action classes, whereas the UCF101 dataset includes 101 classes.

Human Motion Database (HMDB51) is a popular database of human actions. It consists of 51 different action classes extracted from videos captured from YouTube, public databases, and movies. The dataset contains 6766 videos. The diversity

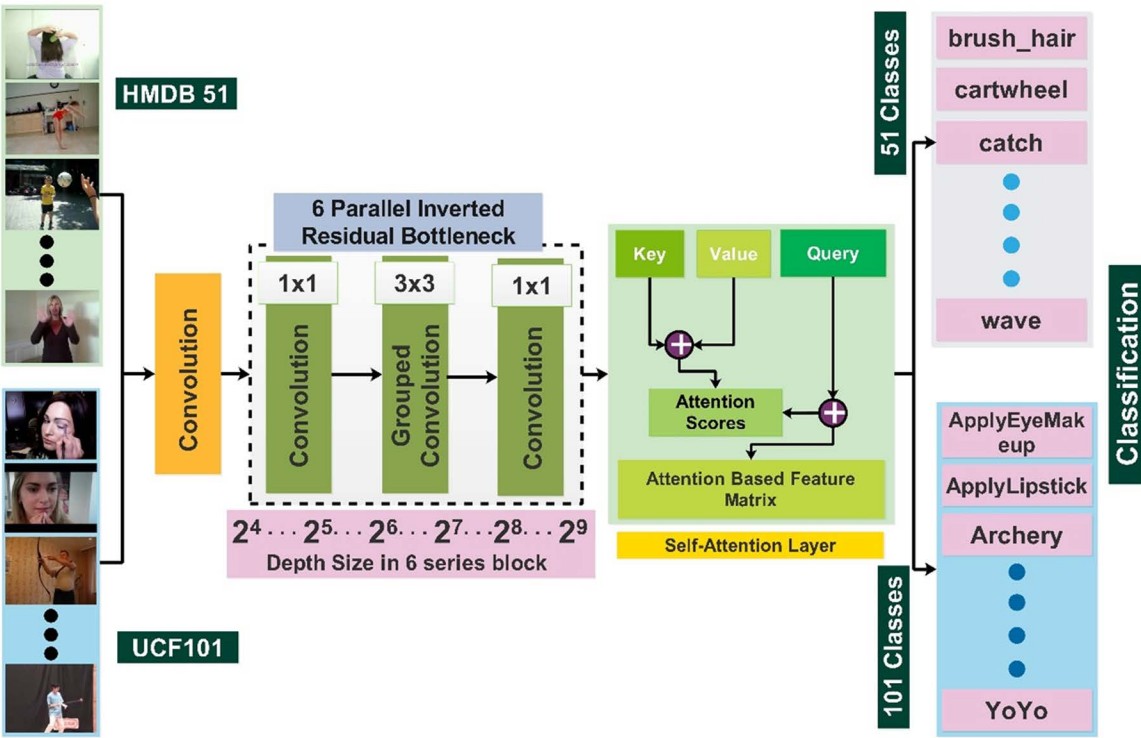

**Fig 3. Proposed architecture for human action recognition on selected action datasets.**

of video sources provides a range of settings and situations for every action class. From these videos, 79133 frames are extracted and provided on the Kaggle dataset repository. Some sample images from different classes are presented in Fig 4.

The UCF101 dataset [40], specifically used for action recognition in videos, comprises 13,320 videos. It is divided into a total of 101 human action classes. The videos have.avi format with an average size of 320×240 pixels. Human-object interaction (i.e., playing guitar, etc.), Body-Motion (i.e., walking), Human-Human Interaction (hugging, etc.), Playing Musical Instruments (i.e., playing the piano, etc.), and Sports (i.e., football or basketball) are some of the categories in this dataset. In this work, the frame extraction uses 3 frames per second. A visual depiction of sample classes of UCF101 is provided in Fig 5.

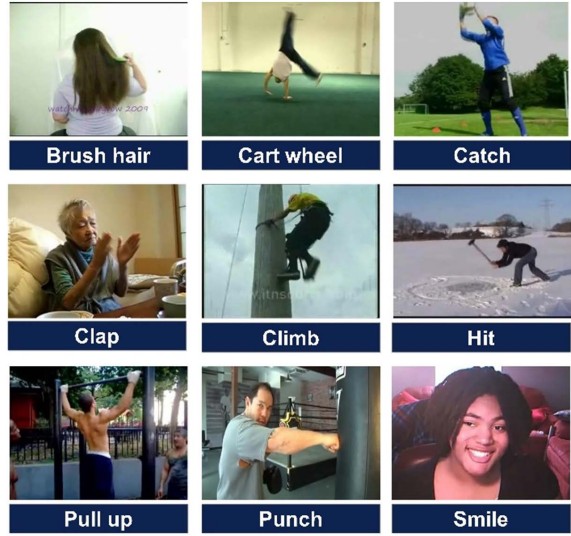

**Fig 4. Sample classes from the HMDB51 dataset.**

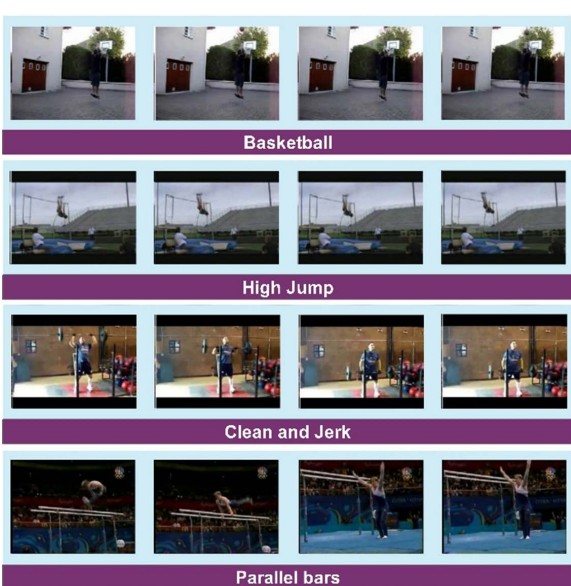

**Fig 5. Sample classes from UCF101 dataset [ 40].**

## 4.2. Proposed inverted residual bottleneck module

In this work, we proposed a new CNN architecture called Inverted Residual Bottleneck Module (InRBM) that consists of a skip connection. The skip connection helps in gradient loss during the backpropagation [41]. InRBM has six parallels. A single InRBM is shown in Fig 6. In this figure, it is shown that the input layer passed to the convolutional layer. The depth size of this layer is 16, the stride is 2, and the filter size is 3x3. After that, the first parallel INRBM block was added, consisting of six paths and one skip connection. The single path consists of five layers in the sequence of convolutional layer, ReLu action, grouped convolutional, ReLu activation, and convolutional layer. The depth size of the convolutional

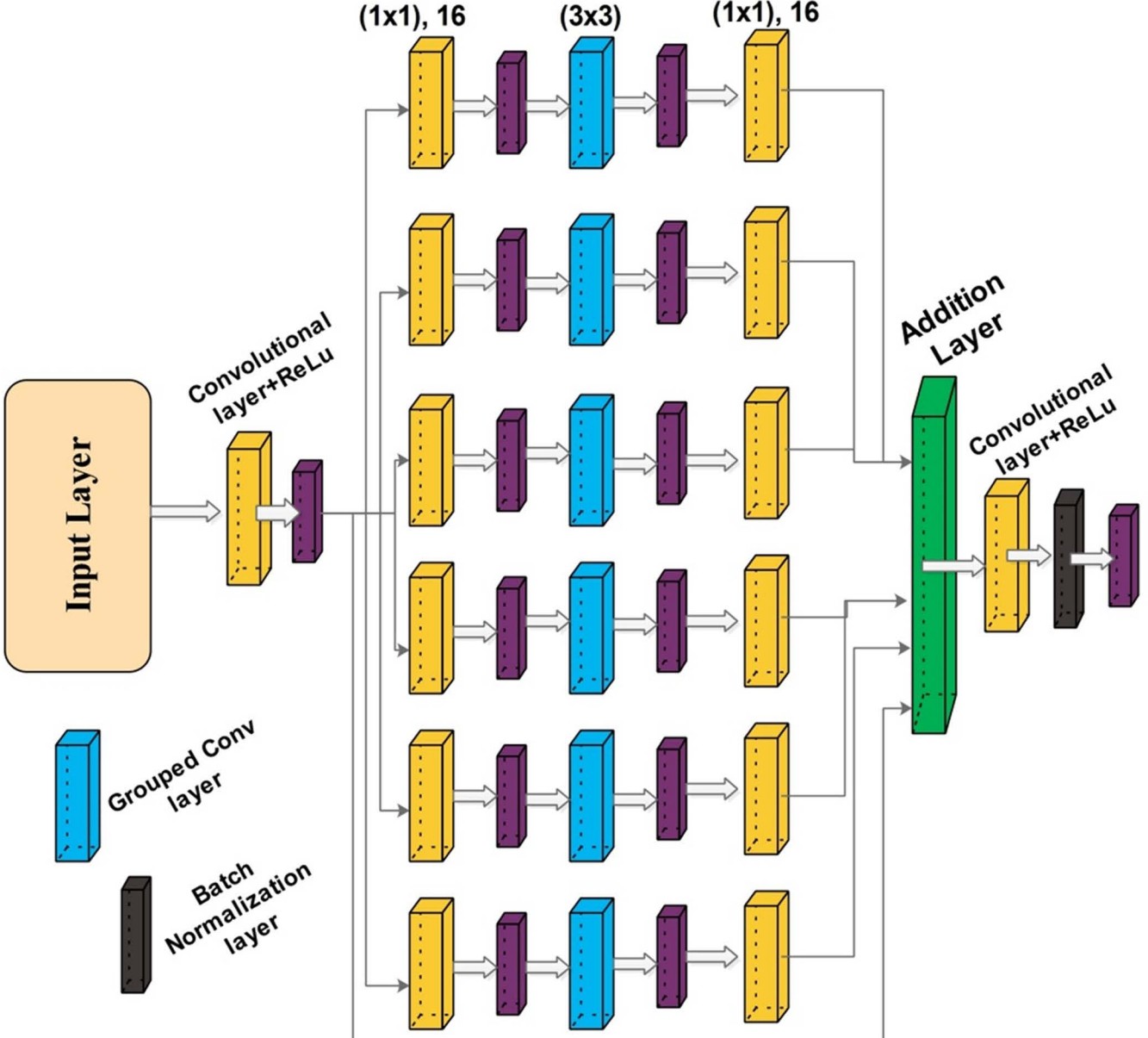

**Fig 6. Proposed inverted residual parallel block.**

layers is 16; strides is one, and the filter size of 1x1. The grouped convolutional layer filter size of 3x3; hence, the sequence of these layers tends to be a bottleneck behavior for convoluting data [42]. The same sequence is considered for this block's rest of the layers. At the end, a skip connection has been added, and the output of all paths is connected in a single addition layer. The addition layer output is passed to the convolutional layer with a depth value of 32 and strides one. A bath normalization layer is also added before the next parallel InRBM block.

The inverted residual structure decreases memory consumption and processing. Employing grouped convolutions drastically lowers the number of parameters compared to ordinary convolutions; hence, efficiency is accomplished [43]. Grouped convolutions are a versatile method for lowering computational costs and parameters, which makes them appropriate for models such as ResNeXt [43] and some iterations of EfficientNet [44]. Skip connections are incorporated directly between bottleneck layers in the architecture, which aids in preserving the gradient flow during backpropagation and is essential for efficiently training deeper networks [45].

The second InBRM block has been added with the same number of layers after the ReLu action layer that follows the batch normalization layer. The depth value in this block increases with $2^k$, where $k = 5$ for block 2. Similarly, the skip connection is added, and all outputs are connected in a single addition layer. This process is continued for the next four parallel blocks, whereas the depth size is increased with $2^k$ and $k = 6, 7, 8, and 9$, respectively. The filter size and stride value remain the same as the first parallel block. In conclusion, bottleneck layers are an essential part of contemporary deep learning architectures because they provide several benefits in terms of computational efficiency, memory utilization, feature representation, and simplicity of training. Together, these advantages improve CNN performance and efficacy across various applications.

## 4.3. Self-attention module

When processing each element, the self-attention mechanism enables the built model to concentrate on distinct segments of the input sequence. It makes it possible for the network to store connections between words or tokens in the sequence, independent of their location. Images of a dataset are represented as a grid of pixels, and the self-attention layer processes each image of a dataset. Three vectors, named value, key, and query, are assigned to each pixel in the image. These vectors are calculated based on learned weight matrices by using linear transformations of the pixel embeddings. The key and value vectors combine information from other pixels and calculate attention ratings, respectively; on the other hand, the query vector represents the representation of the individual pixel. Attention scores of adjacent pixels are calculated, which is accomplished by taking the dot product of a query and the key vector of neighboring pixels, and the result is scaled. The outcome of this procedure is a score that represents the significance or relevance of the second pixel to the first pixel.

The attention scores are normalized using a Softmax function to get attention weights. These weights represent all of the pixels. A pixel with a higher attention weight is more significant in deciding how the current pixel is represented. The weighted sum of the value vectors for each pixel in the image is then calculated using the attention weights. This aggregation process generates a refined and context-aware representation of the input image, in which the weight of each pixel's contribution is determined by its attention weight.

The weighted sum of values functions as the input image's updated representation and is the self-attention layer's final result. In summary, a self-attention layer allows a neural network to focus dynamically on different parts of the input image, effectively capturing spatial dependencies and contextual information. Fig 7 shows the representation of Self-Attention visually.

Mathematically, the self-attention module is defined as follows:

The inputs of the attention module are queries, keys, and values, denoted with $q, k, v$, respectively. These terms are created through a linear transformation as follows:

$$q = UX_q, \tag{6}$$

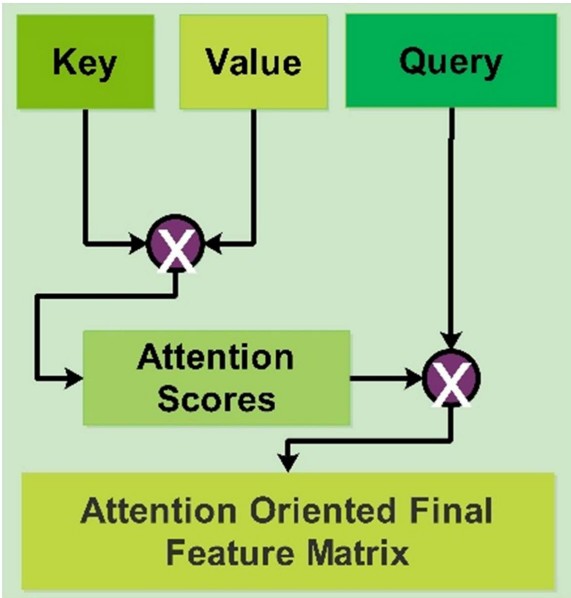

**Fig 7. Self-attention module for the features learning.**

$$k = UX_k,$$ (7)

$$v = UX_v,$$ (8)

Where $U$ denotes the input matrix and $U \in \mathbb{R}^{m \times n}$, the notation $X_q$, $X_k$, and $X_v$ represent weight matrices to be learned. The attention score between $q$ and $k$ is computed using the dot product, which is later scaled down by factor of $\sqrt{d_k}$. This scaling prevents the dot product from growing too large during training. Mathematically, the attention score is computed as follows:

$$attenscore = \frac{qk^T}{\sqrt{d_k}}$$ (9)

The computed score is passed to the softmax function to obtain the attention weights as follows:

$$attenweights = SoftM(attenscore)$$ (10)

Softmax is applied to each row of attention score sum up in one. Hence, the final weighted sum is created using the attention weight as follows:

$$atten(q, k, v) = (attenweights)v$$ (11)

### 4.4. Proposed InBRwSA training

Once the model is designed, the next step is training of the model. In this step, we trained the proposed architecture on the selected datasets. The overall architecture is shown in Fig 8. The optimal hyperparameters are selected for the

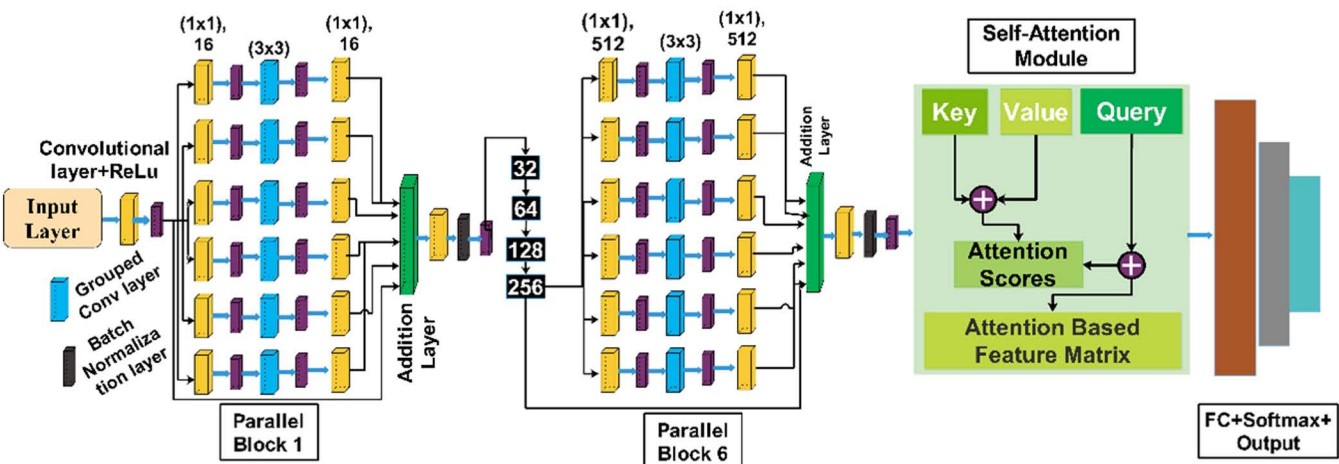

**Fig 8. Complete architecture of proposed inverted bottleneck residual with self-attention CNN.**

proposed architecture for the training process instead of manual initialization. For the initialization of hyperparameters, we utilized the particle swarm optimization algorithm. We passed a range of each parameter to PSO.

After 200 iterations, we obtained the optimal hyperparameters, such as an initial learning rate of 0.00221, momentum value of 0.522, batch size of 64, and ADAM as an optimizer. In addition, we performed a total of 100 epochs in the training phase of the proposed architecture. The trained model is later utilized in the testing phase. The entire architecture in the form of tabular is shown in Fig 9.

### 4.5. Proposed InBRwSA testing

In the testing phase of the proposed architecture, 50% of testing video frames were employed, and features were extracted. The self-attention layer features are extracted that obtained a dimension $N \times 512$. The extracted features are passed to the shallow neural network classifiers for the final action recognition. In this work, we utilized several neural network classifiers, and the shallow wide neural network classifier was selected as the best classifier. The shallow wide neural network classifier consists of two hidden layers, one input layer and one output layer [46]. Visually, the entire testing process is shown in Fig 10. This figure notes that the original video frames are passed to the proposed network and extracted deep features. The extracted deep features are passed to the shallow wide neural network (SWNN) classifier, which returns a prediction-labeled output.

## 5. Results and discussion

### 5.1. Experimental setup

The proposed Inverted Bottleneck Residual with Self-Attention (InBRwSA) architecture is tested, whereas several hyperparameters are employed for the training, as mentioned in the Proposed InBRwSA Training Section. Cross-validation is used with k-folds, and k value is set to 10. The dataset is divided into training and testing at the ratio of 50:50. Maximum epochs are set to 100. The training process takes almost 4 days on both selected datasets. The experiments are performed on MATLAB 2023b with a GPU of NVIDIA GeForce 3060 RTX while having a 12th Gen Intel(R) Core (TM) i5 processor.

| Serial No. | Name | Activations |
|---|---|---|
| 1 | Imageinput 227x227x3 images with "zscore" normalization | 224x224x3x1 |
| 2 | Addition, Element-wise addition of 7 inputs with convolutions of varying depth (6 convolution block + one Residual Block) | 112x112x16x1 |
| 3 | Addition _1, Element-wise addition of 7 inputs with convolutions of varying depth (6 convolution block + one Residual Block) | 56x56x32x1 |
| 4 | Addition _2, Element-wise addition of 7 inputs with convolutions of varying depth (6 convolution block + one Residual Block) | 28x28x64x1 |
| 5 | Addition _36, Element-wise addition of 7 inputs with convolutions of varying depth (6 convolution block + one Residual Block) | 14x14x128x1 |
| 6 | Addition _4, Element-wise addition of 7 inputs with convolutions of varying depth (6 convolution block + one Residual Block) | 7x7x256x1 |
| 7 | Addition _5, Element-wise addition of 7 inputs with convolutions of varying depth (6 convolution block + one Residual Block) | 4x4x512x1 |
| 8 | conv_78, 512 3x3x512 convolutions with stride [2 2] and padding 'same' | 2x2x512x1 |
| 9 | Gapool, 2-D global average pooling | 1x1x512x1 |
| 10 | Flatten, Flatten | 512x1 |
| 11 | Self-attention layer with 512 output channels, 4 heads, 256 key and query channels, and 256 value channels | 512x1 |
| 12 | Fc 10 fully connected layer | - |
| 13 | Softmax | - |
| 14 | Classification | Number of classes for each dataset |

**Fig 9. Detailed, layered architecture of proposed Inverted Bottleneck Residual with Self-Attention (InBRwSA) for action recognition.**

## 5.2. Datasets and performance measures

The experiments are conducted on two different datasets, HMDB51 and UCF101. These datasets are publicly available for research purposes, especially for the HAR. The efficiency and effectiveness of the methodology are tested using different performance measures, such as accuracy, negative rate, and testing computation time. The detailed results are discussed in the sections below.

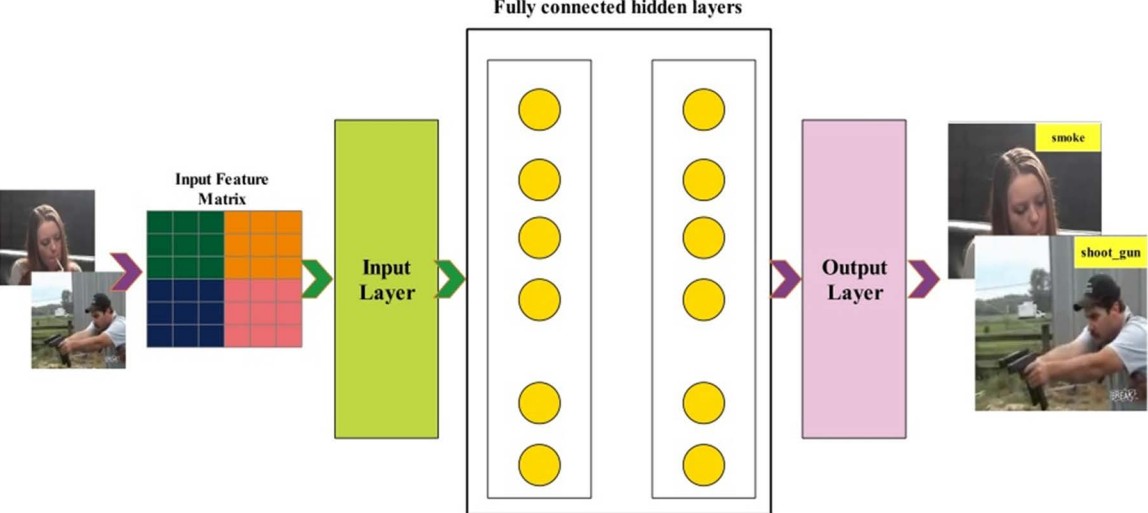

**Fig 10. Proposed architecture testing phase.**

**Table 1. Proposed architecture classification accuracy using the HMDB51 dataset.**

| Classifier | Accuracy (%) | Processing Time (Sec.) | False Negative Rate (%) |
|---|---|---|---|
| SWNN | **78.80** | **134.09** | **21.20** |
| MNN | 59.00 | 463.97 | 41.00 |
| NNN | 42.70 | 408.65 | 57.30 |
| Bi-NN | 40.90 | 441.96 | 59.10 |
| Tri-NN | 38.30 | 462.05 | 61.70 |

### 5.3. Proposed results on HMDB51 dataset

This subsection discusses the proposed architecture results for the HMDB51 action dataset. The proposed results are given in Table 1. The results are presented regarding accuracy, FNR, and testing computation time of the selected classifiers. Table 1 shows the best accuracy of the proposed architecture on the HMDB51 dataset, which is 78.80% for the SWNN classifier, whereas the FNR value is 21.20%. The computation time is also noted, and the obtained time for the SWNN classifier is 134.08 (sec). There are several other listed classifiers in this table that are employed for the classification comparison such as medium neural network (MNN) obtained an accuracy of 59%, narrow neural network (NNN) obtained 42.70%, Bi-layered NN (Bi-NN) obtained an accuracy of 40.90%, and Tri-layered NN (Tri-NN) classifier obtained an accuracy of 38.30%, respectively. These classifiers also consumed more time than the SWNN classifier. Overall, the SWNN classifier shows better classification accuracy than these listed classifiers. The obtained accuracy of the SWNN classifier can be further verified through a confusion matrix illustrated in Fig 11. The diagonal values in this figure show the correct prediction rate.

### 5.4. Classification results on UCF101

This subsection discusses the proposed architecture results using the UCF101 dataset. The UCf101 dataset comprises 101 action classes and several complex and similar classes. Table 2 presents the results of this dataset. In this table, the

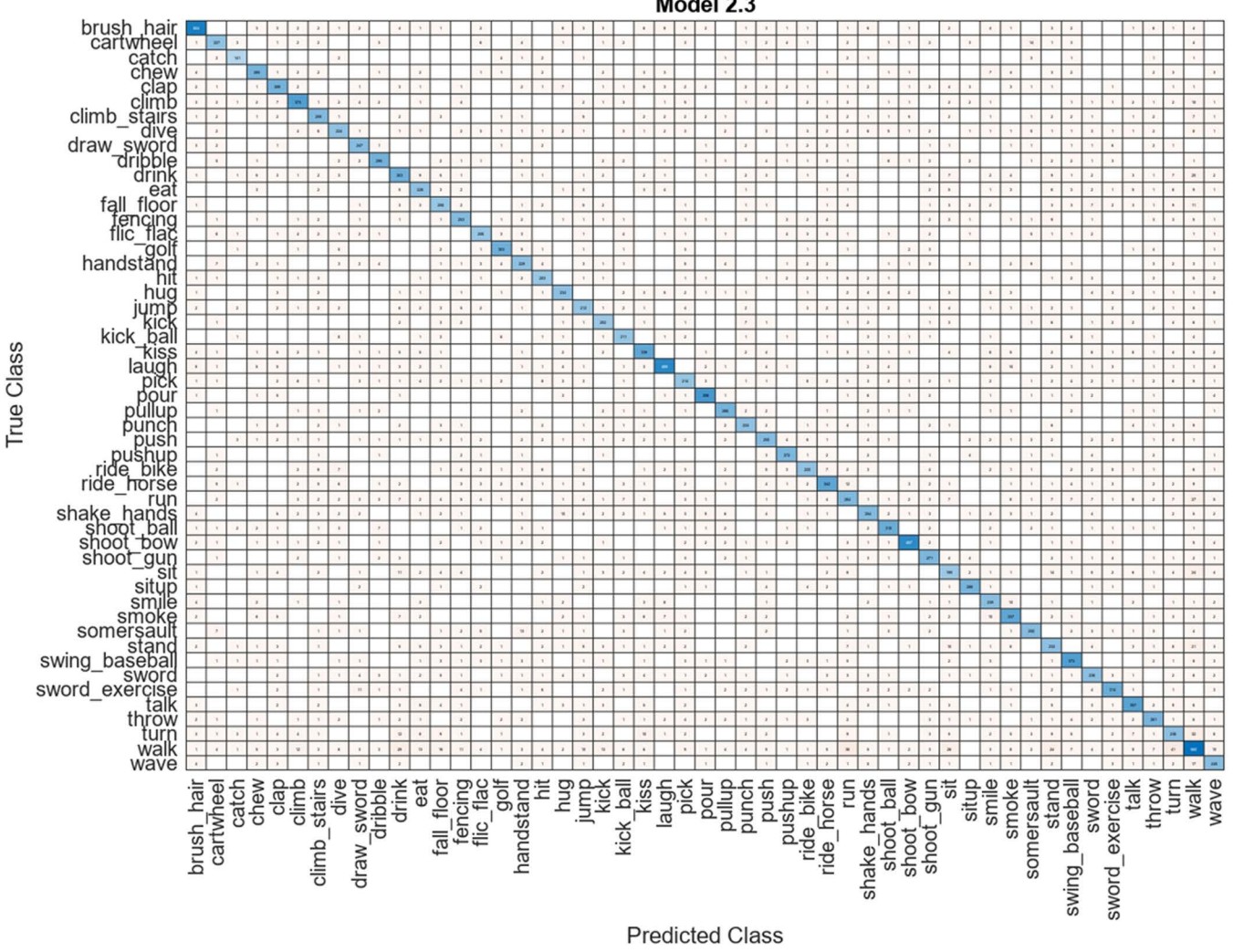

**Fig 11. Confusion matrix for SWNN classifier using proposed architecture.**

**Table 2. Proposed architecture classification accuracy using UCF101 Dataset.**

| Classifier | Accuracy (%) | Processing Time (Sec.) | False Negative Rate (%) |
|---|---|---|---|
| SWNN | **91.80** | **252.10** | **8.10** |
| MNN | 81.40 | 1554.40 | 18.60 |
| NNN | 63.00 | 1387.30 | 37.00 |
| Bi-NN | 61.00 | 1446.90 | 39.00 |
| Tri-NN | 56.10 | 1483.10 | 43.90 |

SWNN classifier obtained the highest accuracy of 91.80%, whereas the FNR value of 8.10. The computation time of this classifier is 252.10 (sec), which is better than the other listed classifies in this table. The classification accuracy of the rest of the classifiers, such as MNN, is 81.40%, NNN is 63%, Bi-NN is 61%, and Tri-NN is 56.10%, respectively. In addition,

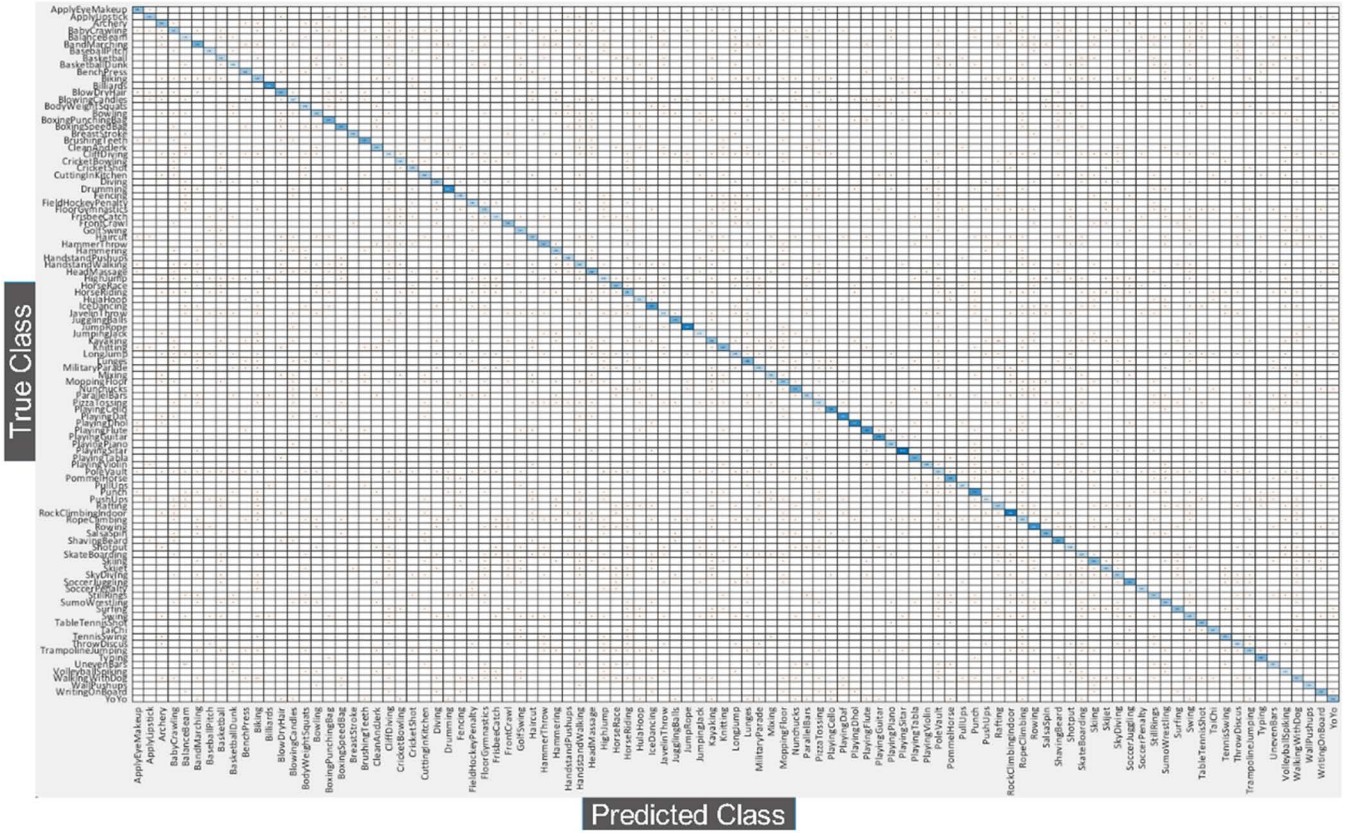

**Fig 12. Confusion matrix for SWNN classifier using proposed architecture using UCF101 dataset.**

the highest time of the other listed classifiers is 1554.40 (sec) for the MNN classifier. Fig 12 shows the confusion matrix of the SWNN classifier for the UCF101 dataset. Hence, the overall SWNN classifier shows the best accuracy using the UCF101 dataset.

### 5.5. Ablation study 1

In the first ablation study, we compared the proposed InBRwSA architecture with state-of-the-art pre-trained models, including AlexNet, GoogleNet, VGG16, VGG19, and ResNet. Comparison is conducted for the selected datasets, such as HMDB51 and UCF101, as shown in Fig 13. In this figure, the first part compares the proposed architecture with pre-trained models using the HMDB51 dataset. The pre-trained models such as AlexNet, GoogleNet, VGG16, VGG19, ResNet18, ResNet50, and ResNet101, and their obtained accuracies are 68.30, 70.14, 65.16, 64.44, 68.90, 71.04, and 74.29%, respectively. The proposed architecture obtained an accuracy of 78.80% for this dataset. The second part of this figure compares proposed and pre-trained CNN architectures for the UCF101 dataset. For this dataset, the pre-trained models obtained accuracies of 78.42, 80.78, 80.36, 79.22, 82.14, 83.29, and 86.24%, respectively. The proposed architecture obtained an accuracy of 91.80%, which is improved than these listed architectures.

 

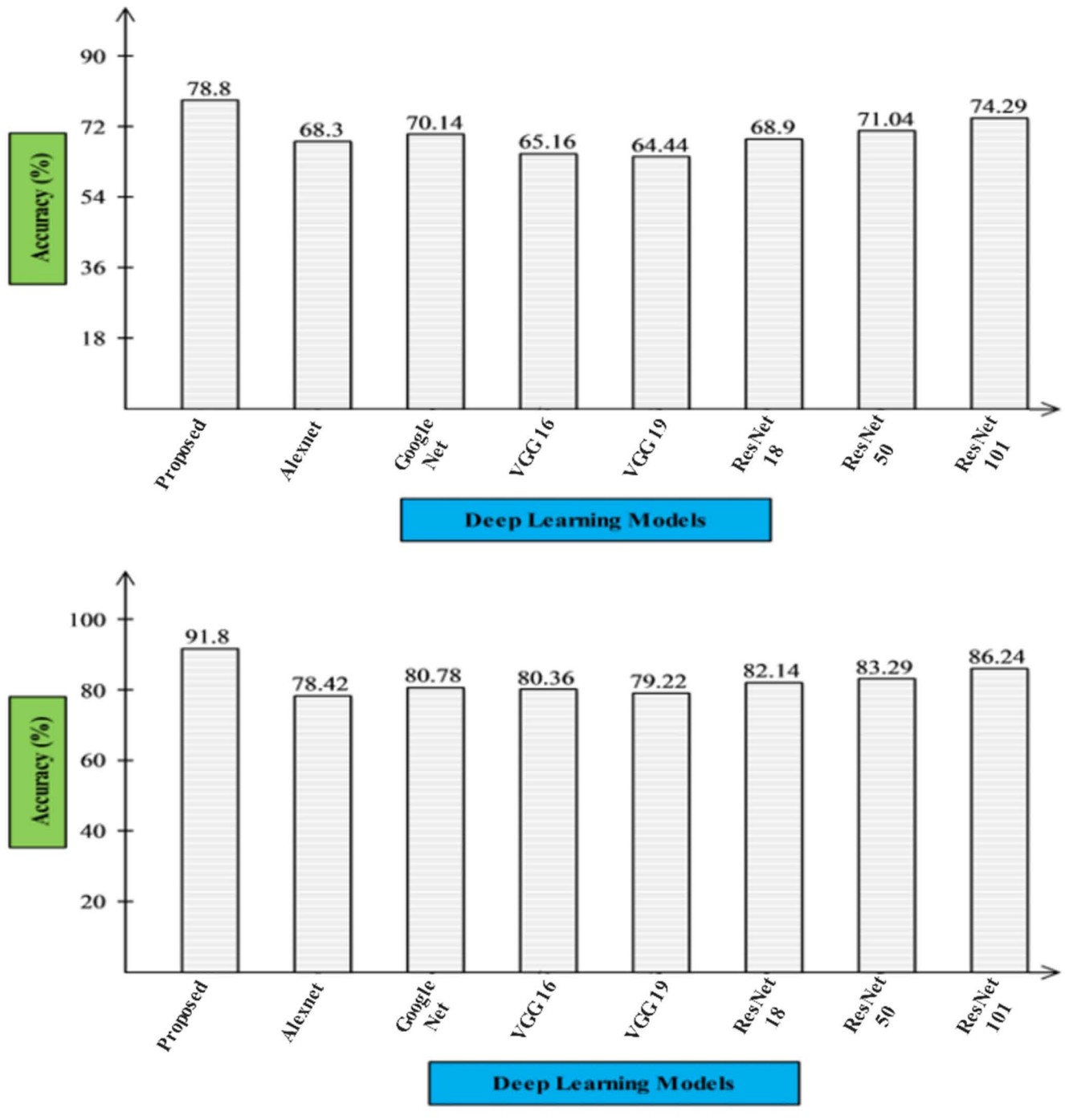

**Fig 13. Ablation study 1- pre-trained deep learning-based comparison for the selected datasets.**

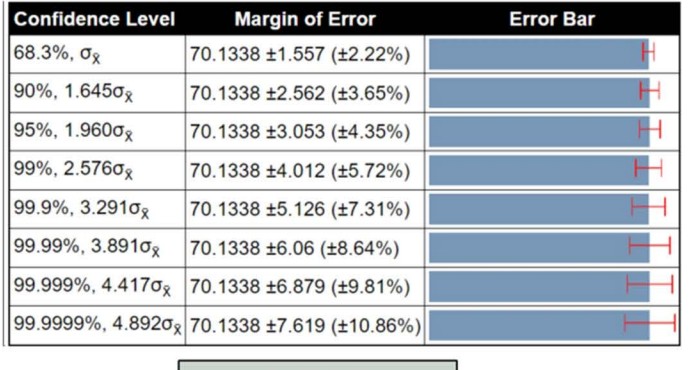

| Confidence Level | Margin of Error | Error Bar |
|---|---|---|
| 68.3%, $\sigma_{\bar{x}}$ | 70.1338 ±1.557 (±2.22%) | |
| 90%, 1.645$\sigma_{\bar{x}}$ | 70.1338 ±2.562 (±3.65%) | |
| 95%, 1.960$\sigma_{\bar{x}}$ | 70.1338 ±3.053 (±4.35%) | |
| 99%, 2.576$\sigma_{\bar{x}}$ | 70.1338 ±4.012 (±5.72%) | |
| 99.9%, 3.291$\sigma_{\bar{x}}$ | 70.1338 ±5.126 (±7.31%) | |
| 99.99%, 3.891$\sigma_{\bar{x}}$ | 70.1338 ±6.06 (±8.64%) | |
| 99.999%, 4.417$\sigma_{\bar{x}}$ | 70.1338 ±6.879 (±9.81%) | |
| 99.9999%, 4.892$\sigma_{\bar{x}}$ | 70.1338 ±7.619 (±10.86%) | |

Analysis for HMDB51 dataset

| Confidence Level | Margin of Error | Error Bar |
|---|---|---|
| 68.3%, $\sigma_{\bar{x}}$ | 82.7813 ±1.455 (±1.76%) | |
| 90%, 1.645$\sigma_{\bar{x}}$ | 82.7813 ±2.394 (±2.89%) | |
| 95%, 1.960$\sigma_{\bar{x}}$ | 82.7813 ±2.852 (±3.45%) | |
| 99%, 2.576$\sigma_{\bar{x}}$ | 82.7813 ±3.749 (±4.53%) | |
| 99.9%, 3.291$\sigma_{\bar{x}}$ | 82.7813 ±4.789 (±5.79%) | |
| 99.99%, 3.891$\sigma_{\bar{x}}$ | 82.7813 ±5.662 (±6.84%) | |
| 99.999%, 4.417$\sigma_{\bar{x}}$ | 82.7813 ±6.428 (±7.76%) | |
| 99.9999%, 4.892$\sigma_{\bar{x}}$ | 82.7813 ±7.119 (±8.60%) | |

Analysis for UCF101 dataset

**Fig 14. Ablation Study 2- Confidence interval-based analysis of proposed architecture on selected datasets.**

**Table 3. Comparison with existing SOTA techniques.**

| Reference | Dataset | Year | Accuracy (%) |
|---|---|---|---|
| Dastbaravardeh et al. [27] | HMDB51 | 2024 | 77.29 |
| Amin et al. [28] | HMDB51 | 2019 | 70.30 |
| Joudaki et al. [29] | HMDB51 | 2024 | 74.28 |
| Abdorreza [30] | HMDB51 | 2023 | 71.13 |
| Yang et al. [31] | HMDB51 | 2022 | 65.90 |
| Hussain et al.[25] | HMDB51 | 2024 | 76.1428 |
| Altaf et al. [22] | HMDB51 | 2024 | 78.6285 |
| **Proposed** | **HMDB51** | **2024** | **78.80** |
| Dave et al. [32] | UCF101 | 2022 | 82.40 |
| Shen et al. [33] | UCF101 | 2021 | 79.60 |
| Ahmad et al. [34] | UCF101 | 2023 | 91.79 |
| Wang et al. [35] | UCF101 | 2024 | 88.60 |
| Korban et al. [49] | UCF101 | 2024 | 85.50 |
| Hanzla et al. [47] | UCF101 | 2024 | 86.90 |
| Zhang et al. [48] | UCF101 | 2024 | 88.10 |
| **Proposed** | **UCF101** | **2024** | **91.80** |

## 5.7. Ablation study 2

In the second ablation study, we conducted a statistical analysis based on the confidence interval and standard error mean (SEM), as illustrated in Fig 14. The analysis is performed based on ablation study 1. For the HMDB51 dataset, all accuracies are selected from Fig 13 and computed with the mean value of 70.13, standard deviation of 4.405, and standard error mean value of 1.557, respectively. Based on these calculated values, we obtained the margin of error value of $70.1338 \pm 3.053\ (\pm 4.35\%)$ for the confidence level $95\%$, $1.960\sigma x$. For the UCF101 dataset, the obtained mean value is 82.78%, whereas the $\sigma = 4.116$. Based on the $\sigma$ and mean value, the standard error mean (SEM) is computed as 1.455, which is later employed for the margin of error (MoE). The MoE computed based on the SEM is 82.7813±2.852 (±3.45%) using a confidence level of 95%, 1.960$\sigma_{\bar{x}}$.

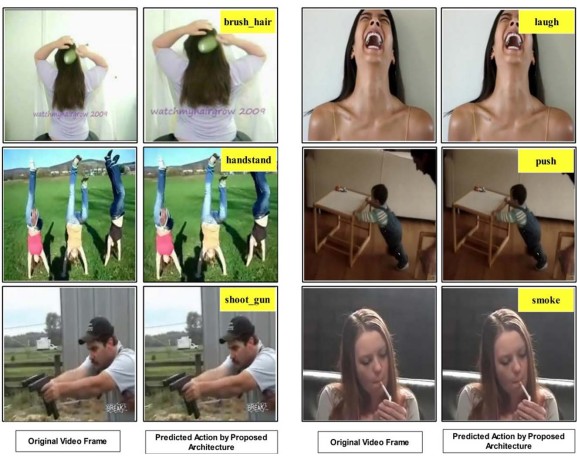

**Fig 15. Proposed architecture prediction results using video sequences.**

**Comparison with state-of-the-art (SOTA) techniques.** In this section, a comparison of the proposed architecture with recent techniques using accuracy value is conducted. Table 3 compares SOTA techniques that employed HMDB51 and UCF101 datasets. In this table, authors in [31] used the HMDB51 dataset and obtained an accuracy of 65.90%. Authors in [28] obtained an accuracy of 70.23% using the HMDB51 dataset. Later on, Joudaki et al. [29] improved the accuracy of the HMDB51 dataset and reached 74.28%. The recent accuracy was further enhanced by Dastbaravardeh et al. [27], who obtained the highest value of 77.29%. In recent techniques proposed by Hussain et al.[25] and Altaf et al. [22] achieved 76.1428% and 78.6285% respectively on HMDB51. The proposed architecture obtained an accuracy of 78.80%, better than the current state-of-the-art techniques. For the UCF101 dataset, Ahmad et al. [34] recently obtained an improved accuracy of 91.79% which was previously 88.60% achieved by Wang et al. [35]. A few recent works by **Hanzla et al.** [47] and Zhang et al. [48] on human action recognition using the UCF 101 dataset reported an accuracy of 86.90 and 88.10, respectively. The proposed architecture obtained improved accuracy on the UCF101 dataset of 91.80%, which shows this work's significance. Finally, some visual prediction results of the proposed architecture are shown in Fig 15. The original images are passed to the proposed architecture in this figure, returning a predicted output.

## 6. Conclusion

Human action recognition has shown enormous attention from computer vision researchers in the last decade. It has employed various data modalities with unique characteristics to accomplish the goal. The recent success of deep learning in the area of machine learning is the best option at present. In this work, we proposed a novel Inverted Bottleneck Residual with Self-Attention architecture for accurate human action recognition. The proposed architecture is based on two modules- i) 6-parallel inverted bottleneck residual blocks and ii) self-attention block. The proposed architecture is designed to bottleneck those, in return, several learning parameters. The hyperparameters of this model are initialized using the PSO algorithm instead of manual selection, which impacts the training accuracy. In addition, the proposed architecture consumes less time, improves accuracy, and is overall more efficient. The experimental process was conducted on two publicly available datasets, HMDB512 and UCF101, and obtained an accuracy of 78.80% and 91.80%, respectively. Overall, the proposed architecture shows improved recognition accuracy. The limitation of this work is a selection of filter size and depth value. The smaller filter size and depth value reduced the number of parameters but, on the other side, impacted the recognition accuracy. In the future, we will consider a fused network that passes two outputs in a single layer to improve current accuracy.

## Author contributions

**Conceptualization:** Yasir Khan Jadoon, Muhammad Attique Khan, Yasir Noman Khalid.

**Data curation:** Muhammad Attique Khan, Yasir Noman Khalid, Jamel Baili.

**Formal analysis:** Yasir Noman Khalid, Nebojsa Bacanin, MinKyung Hong.

**Funding acquisition:** MinKyung Hong, Yunyoung Nam.

**Investigation:** MinKyung Hong.

**Methodology:** Yasir Khan Jadoon, Muhammad Attique Khan, Yasir Noman Khalid, Jamel Baili, Nebojsa Bacanin.

**Project administration:** Yunyoung Nam.

**Resources:** Jamel Baili.

**Software:** Yasir Khan Jadoon, Muhammad Attique Khan, Jamel Baili.

**Supervision:** Yasir Noman Khalid, Yunyoung Nam.

**Validation:** Jamel Baili, Nebojsa Bacanin, MinKyung Hong, Yunyoung Nam.

**Visualization:** Nebojsa Bacanin.

**Writing – original draft:** Yasir Khan Jadoon, Muhammad Attique Khan.

**Writing – review & editing:** Nebojsa Bacanin, MinKyung Hong, Yunyoung Nam.

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
