## [Decision Letter · Decision Letter 0]

17 Jan 2025

Dear Dr. Khan,

Thank you for submitting your manuscript to PLOS ONE. After careful consideration, we feel that it has merit but does not fully meet PLOS ONE’s publication criteria as it currently stands. Therefore, we invite you to submit a revised version of the manuscript that addresses the points raised during the review process.

We look forward to receiving your revised manuscript.

Kind regards,

Feng Ding

Academic Editor

PLOS ONE

Journal Requirements:

5. Thank you for stating the following in your Competing Interests section:  No competing interest

6. Please ensure that you refer to Figure 14 in your text as, if accepted, production will need this reference to link the reader to the figure.

Reviewers' comments:

Reviewer's Responses to Questions

**Comments to the Author**

1. Is the manuscript technically sound, and do the data support the conclusions?

Reviewer #1: Yes

Reviewer #2: Yes

2. Has the statistical analysis been performed appropriately and rigorously?

Reviewer #1: Yes

Reviewer #2: Yes

3. Have the authors made all data underlying the findings in their manuscript fully available?

Reviewer #1: No

Reviewer #2: Yes

4. Is the manuscript presented in an intelligible fashion and written in standard English?

Reviewer #1: Yes

Reviewer #2: Yes

Reviewer #1: 1. In the abstract, the authors state that they selected specific datasets for their model evaluation, but it remains unclear why these datasets were chosen exclusively. Please justify this selection more comprehensively to clarify the model’s applicability and limitations.

2. The abstract should also include the achieved percentage accuracy for each dataset evaluated and report on the time complexity of the proposed model. This addition will enhance the reader’s understanding of the model’s performance and computational efficiency.

3. In the introduction section, the advantages of using a vision-based approach over other modalities, such as sensor-based methods, need to be discussed in more detail. This would help establish the rationale for choosing vision-based methods. Additionally, please cite recent related works to provide a robust context for your study, such as: https://doi.org/10.1109/CVPRW63382.2024.00344;
https://doi.org/10.1109/TII.2024.3431070;
https://doi.org/10.1016/j.aej.2023.11.017

4. The contribution section should be refined to clearly compare your work with these recent papers. Highlighting unique aspects of your approach will strengthen the section.

5. An algorithmic representation of the proposed framework should be included. This will provide readers with a clearer understanding of the method’s step-by-step process.

6. Replace Figure 9 with a concise table that outlines the main module’s input and output.

7. Refine Figure 12 by representing the dataset class names with numbers, as demonstrated in the paper “https://doi.org/10.1109/TBDATA.2024.3489414” This will make the figure more straightforward and consistent.

8. In figure 13 the proposed model is not shown. Please adjust this figure.

9. To further strengthen the discussion, compare your method with recent techniques, and delineate how your approach differs from existing methods. https://doi.org/10.1109/TCE.2024.3373824;
https://doi.org/10.1109/TBDATA.2024.3489414

Reviewer #2: This paper presents a deep learning method for detecting human action. The article is well organized and the proposed network structure is somewhat novel. Here are my comments:

1. The formatting of the formulas throughout the text is very confusing, e.g. some of the formulas are not centered and some are to the left.

2. Figure 8 is very poorly viewed and could be drawn vertically or, instead of indicating the detailed block, replace it with an overall block box and then draw the detailed block separately.

3. Figure 9 is not clearly visible, it should be a clear table, please redraw it.

4. The improvement over the best SOTA is very limited, and both datasets should be reproduced in all comparison methods.

**Do you want your identity to be public for this peer review?** For information about this choice, including consent withdrawal, please see our Privacy Policy

Reviewer #1: No

Reviewer #2: No

---

## [Author Response · Author response to Decision Letter 1]

27 Jan 2025

Dear Respected EiC and honorable reviewers, we revised our manuscript as per the comments and uploaded final version and response sheet in the attachment, thank you

---

## [Decision Letter · Decision Letter 1]

24 Mar 2025

InBRwSANet: Self-Attention based Parallel Inverted Residual Bottleneck Architecture for Human Action Recognition in Smart Cities

PONE-D-24-47480R1

Dear Dr. Khan,

We’re pleased to inform you that your manuscript has been judged scientifically suitable for publication and will be formally accepted for publication once it meets all outstanding technical requirements.

Kind regards,

Feng Ding

Academic Editor

PLOS ONE

Additional Editor Comments (optional):

Reviewers' comments:

Reviewer's Responses to Questions

**Comments to the Author**

Reviewer #1: All comments have been addressed

Reviewer #2: All comments have been addressed

2. Is the manuscript technically sound, and do the data support the conclusions?

Reviewer #1: Yes

Reviewer #2: Yes

3. Has the statistical analysis been performed appropriately and rigorously?

Reviewer #1: Yes

Reviewer #2: Yes

4. Have the authors made all data underlying the findings in their manuscript fully available?

Reviewer #1: Yes

Reviewer #2: Yes

5. Is the manuscript presented in an intelligible fashion and written in standard English?

Reviewer #1: Yes

Reviewer #2: Yes

Reviewer #1: (No Response)

Reviewer #2: (No Response)

**Do you want your identity to be public for this peer review?** For information about this choice, including consent withdrawal, please see our Privacy Policy

Reviewer #1: No

Reviewer #2: No

---

## [Editor Report · Acceptance letter]

PONE-D-24-47480R1

PLOS ONE

Dear Dr. Khan,

I'm pleased to inform you that your manuscript has been deemed suitable for publication in PLOS ONE. Congratulations! Your manuscript is now being handed over to our production team.

Kind regards,

on behalf of

Dr. Feng Ding

Academic Editor

PLOS ONE